# A Delphi Study to Develop Items for a New Tool for Measuring Child Neglect for Use by Multi-Agency Practitioners in the UK

**Simon Haworth** [1,*] **, Paul Montgomery** [2] **and Jason Schaub** [1]

1   Department of Social Work & Social Care, School of Social Policy, University of Birmingham, Birmingham B15 2TT, UK
2   Department of Social Policy, Sociology and Criminology, University of Birmingham, Birmingham B15 2TT, UK
*   Correspondence: s.p.c.haworth@bham.ac.uk

**Abstract:** Social work and allied professions can struggle to accurately assess child neglect. Our research project is developing a new child neglect measurement tool for use by multi-agencies to address this issue. Phase two of this project employed a Delphi study to gather the views of a range of experts to help develop it. There were two important stages to inform the Delphi study: a systematic review of child neglect measures, and three online focus groups with a purposive sample of 16 participants with expertise in child neglect (academics, practitioners, and experts by experience). We then conducted a three-round modified online Delphi study with a purposive sample of 60 international panellists with expertise in child neglect. We followed the CREDES guidelines for the rigorous application of the Delphi technique. The panel generated salient items for the tool and scaled these for importance. The panel reached consensus for 18 items and 15 elements for the tool. The items included neglect type, chronicity, and severity. The elements included hyperlinks to research and the use of 10-point scales. The draft tool is short and may be useable by a range of practitioners in multi-agency settings. It is inclusive of social harms, such as poverty and social isolation. It will now be piloted.

**Keywords:** child neglect; measurement; assessment; Delphi study; social work; social harm





## 1. Introduction

### 1.1. Child Neglect and Its Complex Nature

Child neglect is prevalent across all societies and its impacts and costs for children, families, communities, and societies suggest it merits a more rigorous and complete research evidence base (Daniel et al. 2010; Dubowitz 2007; Mulder et al. 2018). Neglect accounted for 52% of initial child protection plans in England during 2020–2021 (Department for Education 2021). Similarly high levels of neglect coming to the attention of statutory services can be found in countries such as the USA, Canada, and the Netherlands (Euser et al. 2010; Stoltenborgh et al. 2015). In the USA, 75% of initial referrals to child protective services are for neglect, as are the majority of recurrent maltreatment reports (Jonson-Reid et al. 2019; US Department of Health and Human Services 2021).

Child neglect is complex and has varying presentations from mild to severe, and episodic to chronic (English et al. 2005). It can feature a range of interlinked issues from personal through societal levels, including variable levels of care, problematic parent-child relationships, breakdowns in social relationships, neighbourhood deprivation, and a wide range of social harms (Chambers and Potter 2009; Dufour et al. 2008; Lacharité 2014; Shanahan et al. 2017). Of all forms of maltreatment, neglect can lead to some of the most damaging long-term impacts on development, wellbeing, and behaviour (Daniel 2015; Stevenson 2007). It is important to note that the impacts of neglect can be not just harmful but fatal (Sidebotham et al. 2016).

There is a significant range of definitions of child neglect from research, government, and practice (English et al. 2005). Definitions vary among countries and, indeed, among

states and jurisdictions within countries (Horwath 2013). There is also a range of conceptual models and typologies of child neglect (Horwath 2007; Sullivan 2000). These issues create a complex picture for assessment.

### 1.2. Assessment Challenges

The assessment of neglect raises significant challenges for social work and allied professions, such as health and education. These assessments can be filled with ambiguity because neglect is both opaque and complex (Brandon et al. 2009; Doherty 2017; Stewart et al. 2015). Further, the involvement of children's social work and allied professions is principally based on community and social constructions of neglectful care rather than empirical evidence on what harms children (Dubowitz et al. 2005; Munro 2020). This is largely the case across the world (Dubowitz and Merrick 2010; Horwath 2013).

There has been limited rigorous research into the assessment and measurement of neglect, with no gold standard for its measurement (Bailhache et al. 2013; Haworth et al. 2022; Horwath 2013; Morrongiello and Cox 2020). Rigorously developed and tested evidence-based measurement tools and frameworks are important for accurately measuring child maltreatment (Bailhache et al. 2013; Parker 2020), and can support balanced, systematic, and analytical assessments (Barlow et al. 2010; White and Walsh 2006). In the absence of clear standards and effective tools, practitioners can tend to rely on practice wisdom and subjective judgments (Hines et al. 2006; Stewart et al. 2015; Stokes and Taylor 2014), looking to reduce complex assessments and decisions to manageable decision-making strategies (Broadhurst et al. 2010; Cummins 2018; Platt and Turney 2014). The multi-agency context for identifying and addressing neglect can itself pose complications for an effective assessment of the issue (Thompson 2016). Health, early help, and education agencies are commonly involved in assessing and responding to child neglect (Sharley 2020).

Research has highlighted the varying and varied standards of decision-making within child protection and its impacts (Barlow et al. 2010; Dorsey et al. 2008). Poor, inadequate, or incomplete assessments play a substantive role in significant harm and/or fatality from neglect (Brandon et al. 2020). They can lead to delay, drift, and error in professional decision-making and actions (Helm 2010). In the UK, Ofsted's (Office for Standards in Education 2014) thematic inspection of responses to neglect deemed assessments to be of heterogenous standards, with 50% of assessments considered inadequate. Within this context of assessment challenges, neglect was an issue in 68% of fatal cases and 83% of non-fatal harm cases in the 368 serious case reviews carried out into children who have died or been seriously harmed through abuse or neglect in the UK between 2014 and 2017 (Brandon et al. 2020). Of the total 1750 maltreatment deaths in the USA, 1277 (73%) were due to neglect (US Department of Health and Human Services 2022).

### 1.3. The Research Project

The overarching aims of this research project are to develop a valid, simple, and practitioner-accessible multi-agency child neglect measurement tool, titled the 'family and wider social neglect measurement tool', to support evidence-based and informed assessments that are also inclusive of key social harms, such as poverty and community deprivation. It consists of three phases:

- Phase one (completed) was a systematic review of national and international, clinical and academic, and single index and multi-dimensional measures of child neglect.
- Phase two, presented here, was an online Delphi study (conducted with a participating local authority in Wales).
- Phase three will pilot the new draft child neglect measurement tool with the participating local authorities, their partner agencies (including health and education), and linked third-sector organisations.

This is a collaborative project, with significant engagement with practitioners and experts by experience (parents with experience of professionals intervening for (suspected)

child neglect). This should promote the research's practice relevance and ensure that social work values are mobilised (Campbell et al. 2017; Uttley and Montgomery 2017).

Our child neglect theory of change (see Figure 1 below) provides a framework to guide the project. It was developed from a review of the literature on neglect (including its key dimensions and drivers) and the literature on children's needs, alongside consultation with our advisory group. It depicts the neglect typology used here and includes key risk and protective factors at personal, family, professional, community, and societal levels. It aims to simply capture the complex social mechanisms involved in neglect.

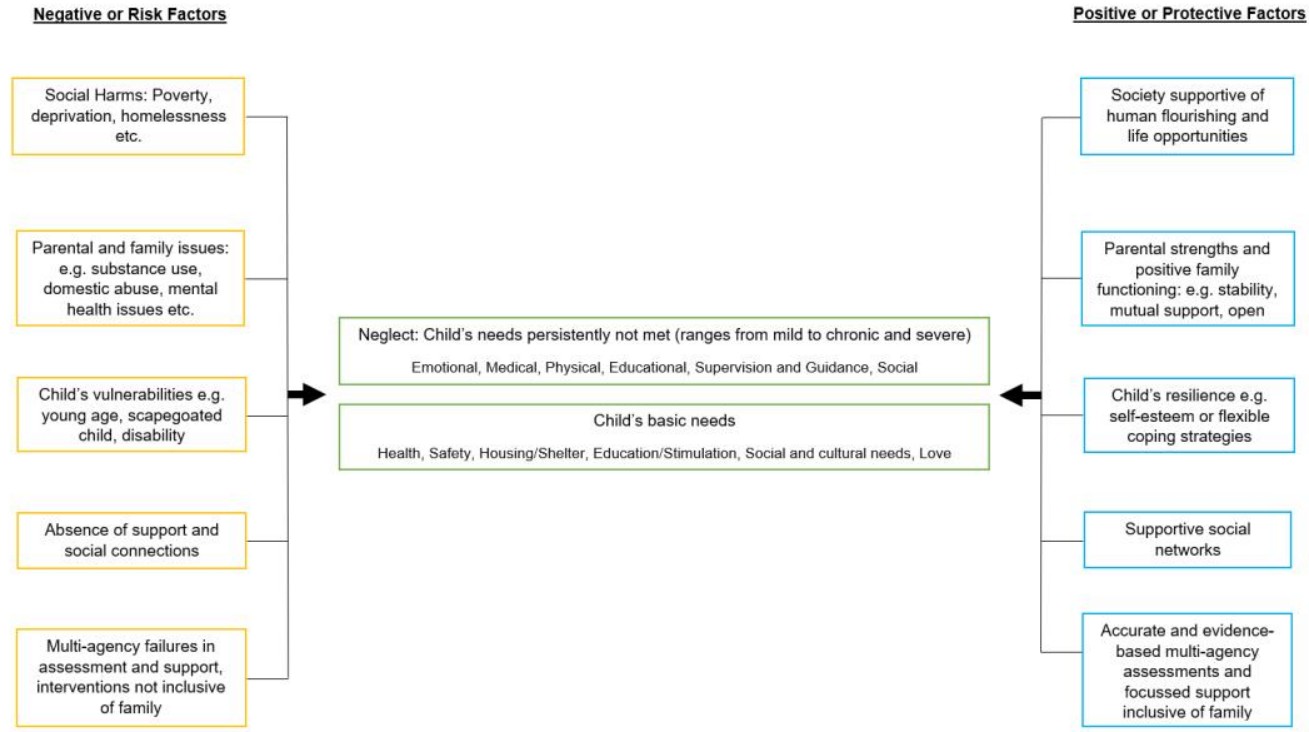

**Figure 1.** Child neglect theory of change.

The social harm approach has informed this project. It recognises that individuals are harmed through the non-fulfilment of their needs and the denial of social resources to exercise life choices within deeply unequal societies (Pemberton 2016). The relationships among poverty, a range of socioeconomic disadvantages, and neglect are well established, if complicated (Bywaters et al. 2016; Carter and Myers 2007; Shanahan et al. 2017). The adoption of a social harm framework can support the understanding and assessment of neglect to move from a reductive vision of harm caused solely by parents to one that recognises and appreciates the range of relational, social, and structural causal and contributing factors present in neglect cases (Lacharité 2014).

This paper reports the Delphi study phase of the project. This was employed to develop items and elements for the draft tool, building on the findings of the preceding systematic review within the overarching evidence-based project. It offered a systematic and efficient approach to gathering the views of a range of experts (Khodyakov et al. 2020).

## 2. Methods

The Delphi method is suited to explore areas where controversy, complexity, debate, or limited empirical evidence exist (Linstone and Turoff 2002; Smart and Grant 2021), as is the case for child neglect and its measurement (Daniel et al. 2010; Dubowitz et al. 2005; Morrongiello and Cox 2020). Delphi studies use a series of discussions or surveys to explore consensus on disputed topics (Linstone and Turoff 2011). We conducted an online modified Delphi study to gather the views of a range of experts to help develop the

new measurement tool. The Delphi was modified through the inclusion of a discussion board (set up via Padlet) to encourage active discussion between rounds (Khodyakov et al. 2020). Such studies offer opportunities for the systematic but also convenient and efficient engagement of relatively large numbers of geographically distributed key stakeholders (Grant et al. 2021) but have the potential pitfall of lower levels of panellist engagement (Khodyakov et al. 2016). We wrote (a priori) and followed a protocol for the Delphi study.

To inform the Delphi study, we first undertook a systematic review of measures of child neglect and then conducted three online focus groups, as described below. As Khodyakov et al. (2016) suggested, 'The Delphi method complements the results of systematic evidence reviews with consensus-focused engagement of experts and stakeholders in emerging areas where there is a lack of rigorous research or where consensus is needed on how to apply research findings . . . ' (p. 354).

Ethics approval was sought and received through the University of Birmingham (ERN_21-0041). Ethical awareness was maintained at all stages, including the full consideration of the participants' wellbeing before, during, and after their engagement (Butler 2002). A clear description of the purpose and processes of the research was provided to the participants as part of their engagement. Voluntary consent was provided by all participants. The data from all the stages were anonymised and stored securely. Figure 2 depicts the stages of this study.

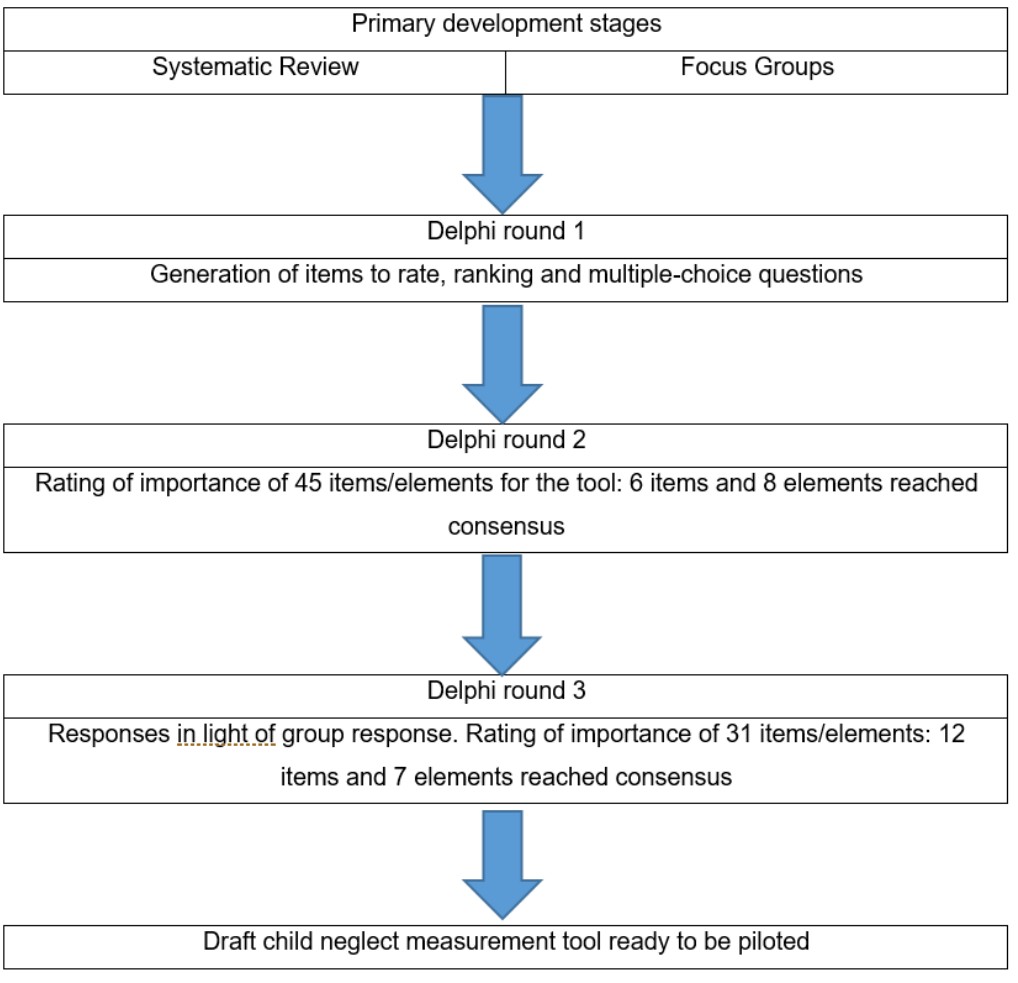

**Figure 2.** Flow chart of stages in the Delphi study.

### 2.1. Study Participants

Panellists can be considered the lynchpin of Delphi studies (Green et al. 1999; Fish and Busby 2005). They need to provide a depth and breadth of knowledge on the topic under

investigation (Linstone and Turoff 2011; Hamlet et al. 2018). We set clear inclusion criteria for an international panel of experts in child neglect (primarily from the UK and the US), with diverse views on the subject through either personal or professional experience. The purposive sample was recruited from these eligible groups:

- Researchers in the field of child neglect;
- Researchers in the field of measurement in social work;
- Multi-agency practitioners who work with child neglect, including frontline workers, those based in learning and development teams, senior practitioners, and managers;
- Experts by experience—parents with experience with professionals intervening for (suspected) child neglect.

We recruited for the focus groups and Delphi panel through the participating local authority, our existing networks, and contacting the authors of key texts in the fields of measurement tools in social work and neglect. We employed snowball sampling for academics and experts by experience, where those recruited were asked to suggest others with relevant specialist knowledge (Montgomery et al. 2019). All experts by experience were spoken with individually to ensure fully informed consent. All focus group members were invited to take part in the surveys. We emailed each potential participant for the focus groups and Delphi rounds between October 2021 and March 2022.

### 2.2. Primary Development Stages

#### 2.2.1. Systematic Review

We undertook a systematic review of national and international, clinical and academic, and single index and multi-dimensional measures of child neglect (Haworth et al. 2022). The review found a distinct lack of evidence-based, valid, or reliable child neglect measurement tools. Only four studies, all from North America, met the inclusion criteria and the gold standard of an assessment by a qualified children's social worker or assessor working within children's social work. Only one tool, the Child Neglect Index (Trocmé 1996), was considered feasible for practice, with the modifications of the Modified Maltreatment Classification System considered too complicated and cumbersome in both our review and the study of Dubowitz et al. (2005) examining the tool. Analysis revealed that although the included tools had strengths, they excluded some key features of neglect, including neglect chronicity and the range of factors that can contribute to neglect occurring, including social harms. Studies of 'popular' tools, such as the Graded Care Profile 2 and HOME, have lacked methodological rigour and have not been assessed against the gold standard of a contemporaneous assessment by a qualified children's social worker or by an assessor working within children's social work (Haworth et al. 2022). The review recommended that child neglect measurement tools need to be robustly tested in social work settings to satisfy the criteria of validity, reliability, and practice/clinical utility.

#### 2.2.2. Online Focus Groups

Synchronous online focus groups can be as effective in gaining information from participants as face-to-face groups (Abrams and Gaiser 2017), but with the advantages of reducing logistical issues and the ease of recording and transcription (Cher Ping and Chee 2001). We facilitated three synchronous online focus groups with practitioners, academics, and experts by experience in February/March 2022 to build on the findings of the systematic review, generate first-round items, and better understand a range of views on what was needed in our new measurement tool. The participants were provided with a summary of the findings of the systematic review to read and reflect on prior to engaging with the focus group.

One focus group constituted experts by experience and two professionals and academics. We view practitioners as our primary 'users', as they will be using the measurement tool. The approach adopted was attentive to the potential for participants to feel pressure to conform to dominant views and socially acceptable identities and avoided potentially mixing people with opposing interests (Green 2009). We produced separate information

and consent forms for experts by experience and practitioners/academics, with attention given to the accessibility of the language.

### 2.3. Online Modified Delphi Study

We conducted a modified online Delphi, involving three anonymous sequential surveys administered through Qualtrics between April and July 2022. All surveys were piloted with two experts by experience, two practitioners, and two academics prior to being administered to encourage the development of robust, clear, and comprehensible questions (Barrington et al. 2021). Each round remained open for 2 weeks.

We followed the CREDES guidelines (Jünger et al. 2017) for the systematic and rigorous application of the Delphi method. Given that the quality of the results and recommendations ' . . . largely depends on the rigour of the application' (Jünger et al. 2017, p. 703), we applied the Delphi technique systematically and rigorously and demonstrated transparency and clarity in the methodological decisions. Defining participant consensus prior to the commencement of the study was essential (Grant et al. 2018; Jünger et al. 2017). We predetermined that the Delphi would stop after three rounds. Key Delphi study experts describe this predetermined approach as good practice because it reduces many forms of bias (Chaffin and Talley 1980; Linstone and Turoff 2011).

Panellists without prior experience of the Delphi process can experience difficulty understanding the processes involved and engaging meaningfully (Biggane et al. 2019). We proactively maintained contact with panellists and provided clear self-explanatory instructions (including a short video on the essential elements of Delphi studies and how to participate online) for less experienced panellists (Beretta 1996; Khodyakov et al. 2020). To facilitate participation, we ensured each survey did not take longer than 30 min to complete (Donohoe et al. 2012).

### The Delphi Rounds

Round one of the Delphi study was an open survey. Panellists were asked to consider and generate salient items for the tool. They were also asked ranking and multiple-choice questions to start to narrow down some of the very broad ideas from the focus groups on what should be in the tool. The panel rated 45 items (distinct parts for the tool that constitute what the tool assesses and focusses on, for example, a scale for neglect severity and the neglect definition used) and elements (features of the tool's design and look that support its aims, for example, hyperlinks for research and the use of 10-point scales) for the tool in round two on 9-point Likert scales. The following criteria were applied:

- Scores of 1–3 indicated that an item was of limited importance for the tool;
- Scores of 4–6 indicated that an item was important but not essential for the tool;
- Scores of 7–9 indicated that an item was critically important for the tool.

Panellists were also asked to comment on the reasoning for their ratings in free-text boxes located beneath scales. The survey for round three modified that of round two through the inclusion of group statistical responses, asking panellists to re-evaluate their responses in light of this information. The panel rated 31 items/elements in round three. We provided panellists with controlled feedback in the form of summaries of responses (de Meyrick 2003). A range of studies using mixed panels of experts have found that consensus is most likely to be achieved by providing summary feedback to all panellists (as opposed to feedback for each different stakeholder group) and providing the rationale behind the responses (Brookes et al. 2016; Fish et al. 2018; Meijering and Tobi 2016). We applied this approach. The panellists were provided with simple colour-coded feedback (based on the Ram analysis technique)—green indicated it was rated as essential, yellow indicated it was important but not essential, and red indicated it was of limited importance (Grant et al. 2021; Montgomery et al. 2019). They were also provided with the basic average rating for each item by the whole panel. The steps taken led to consensus on the items to include in the neglect measurement tool. The facilitation of an online discussion board between the Delphi rounds encouraged active discussions among the panellists (Khodyakov et al. 2020).

*2.4. Analysis*

2.4.1. Qualitative Analysis

We analysed the data from the focus groups using manual thematic analysis, as the data set was relatively small (Braun and Clarke 2019). The manual method implemented allowed for a deep understanding of the material and reflection on some of the nuances in both the meaning and language used by the range of participants (Sykora et al. 2020). In order to improve internal validity, we undertook two primary steps. First, two members of the research team independently coded and analysed the same focus group transcript to compare the findings and interpretations (Bird et al. 2013). Second, we checked with two participants from each focus group that the themes generated seemed reasonable to their experience (Elliott et al. 1999).

The qualitative data gathered through the Delphi rounds were in the form of short-form free-text data answers. We analysed these data using qualitative content analysis. This approach emphasises the construction of meaning from the data, so the categories were not pre-decided; rather, they emerged from the data (Goodings et al. 2013; Snee 2013). As the data set was relatively small, manual coding was undertaken, supported by the Qualtrics platform to count the categories that emerged from the data (Chew and Eysenbach 2010). Those most frequently present were then taken forward as the key concepts for the panellists to consider in the next round.

2.4.2. Quantitative Analysis

We analysed the rating and ranking data to determine the existence of consensus among the participants (Grant et al. 2021; Khodyakov et al. 2020). Lynn (1986) suggested that for a tool to achieve content validity, a minimum of 80% of experts should agree on each item. This threshold has been applied in studies by Eubank et al. (2016) and Paek et al. (2018), for example. The following consensus definition was applied in this study:

> *Consensus will be achieved when 80% or greater of participants rate an item as of critical importance, so 7, 8, or 9 on the 9-point Likert scale.*

We analysed the multiple-choice data from round one through simple multiple-response analysis on the Qualtrics platform. We analysed both percentages for each option and interquartile ranges to assess consensus (Beiderbeck et al. 2021). The options with higher percentages progressed to round two, with the cut-off point set where the percentage decreased significantly from one option to the next, signifying the option as a significantly less popular choice. We analysed the ranking data from round one by calculating the mean scores, with the cut-off point set where the mean increased significantly. The lower the mean score, the higher the panel ranked that item. Figure 3 depicts this analysis stage.

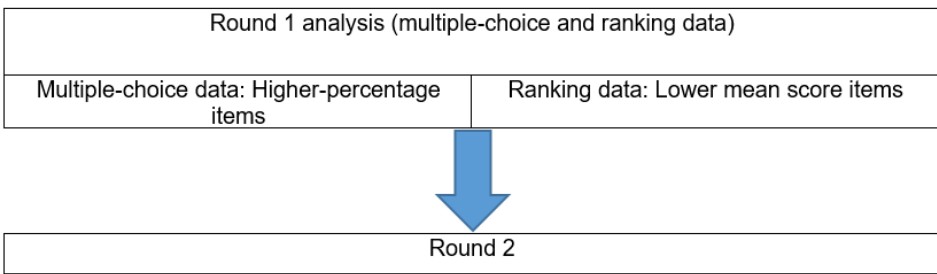

**Figure 3.** Round one analysis of multiple-choice and ranking data.

We applied the consensus definition to data gathered in rounds two and three to determine which items to include in the tool. Items were included in the tool when 80% of the panellists or more rated them as critically important and where at least two out of the three different expert groups accorded this rating. We, therefore, carried out an analysis of variance among the three expert groups. As the data were non-parametric, we applied the

Kruskal–Wallis one-way analysis of variance to see whether the median responses for each item differed significantly by expert group (Hohmann et al. 2018).

## 3. Results

### 3.1. Focus Groups

We invited 16 experts to participate in the focus groups, and all agreed to participate. As the groups had an analytical function, smaller numbers of participants were desirable (Acocella and Cataldi 2020). Seven of the participants were experts by experience, seven were practitioners (from social work, health, education, and family support fields), and two were academics. The majority were white (14), with one participant of mixed ethnicity, and one Asian. Table 1 shows the sociodemographic characteristics of the focus group participants.

**Table 1.** Sociodemographic characteristics of the focus group participants.

| Total (*n* = 16) | Practitioners/Academics (*n* = 9) | Experts by Experience (*n* = 7) |
|:---:|:---:|:---:|
| Sex | | |
| Male | 4 | 1 |
| Female | 5 | 6 |
| Age (years) | | |
| 18–39 | 2 | 4 |
| 40–59 | 6 | 3 |
| Over 60 | 1 | |
| Ethnic group | | |
| White | 9 | 5 |
| Mixed/multiple ethnic group | | 1 |
| Asian/Asian British | | 1 |
| Professional role | | |
| Academic | 2 | |
| Social worker | 2 | N/A |
| Manager | 2 | |
| Other professional | 3 | |

The focus groups ran for up to 60 min to ensure that a range of topics was covered but that participants did not become fatigued. Clarity on the topics to be discussed and proposed timings for each topic supported them to run smoothly (Bloor et al. 2001). The groups provided us with important perspectives on measuring neglect and a more focussed survey for round one (Kvale and Brinkmann 2015; Keeney et al. 2001). They supported the understanding of the language and the concepts the three different expert groups used on child neglect. This was important to ensure that we used language in the study that was understandable and relevant for all (Barrington et al. 2021).

### 3.2. Delphi Study

We recruited 75 Delphi panellists, with a view to accepting a response rate of 50, as attrition is a feature of Delphi studies. Sixty (80%) agreed to participate. The number recruited was slightly higher than longstanding views on the desired numbers of panellists for Delphis and more recent reports on desired numbers for online Delphis (Khodyakov et al. 2020; Linstone and Turoff 2002). This decision was taken to ensure the inclusion of sufficient numbers from each expert group and to ensure the participating local authority had sufficient multi-agency representation. The majority of the panellists identified as white (83%) and were in professional roles (70%). Academics comprised 17% of the panel; experts by experience, 13%. The completion rates were very high: 90% for the academics, 88% for the professionals, and 87.5% for the experts by experience. Table 2 shows the characteristics of the Delphi panel and their completion rates.

**Table 2.** Characteristics of the Delphi panel and completion rates.

| | Total (*n* = 60) | Practitioners (*n* = 42) | Academics (*n* = 10) | Experts by Experience (*n* = 8) |
|---|---|---|---|---|
| Age (years) | | | | |
| 18–39 | 14 | 1 | 5 | |
| 40–59 | 25 | 6 | 2 | |
| Over 60 | 3 | 3 | 1 | |
| Ethnic group | | | | |
| White | 38 | 8 | 6 | |
| Mixed/multiple ethnic group | 2 | 1 | 1 | |
| Asian/Asian British | | | 1 | |
| Other ethnic group | 2 | 1 | | |
| Professional role | | | | |
| Social worker | 17 | N/A | N/A | |
| Manager | 8 | | | |
| Other professional | 17 | | | |
| Completion rate % | 88% | 90% | 87.5% | |

Completion rate is defined in our study as completing the 3 Delphi rounds.

Table 3 shows the items (distinct parts of the tool that constitute what the tool assesses and focusses on, for example, a scale for neglect severity and the neglect definition used) and elements (features of the tool's design and look that support its aims, for example, hyperlinks for research and the use of 10-point scales) that reached consensus to be included in the draft tool. Eighteen items reached the consensus threshold in total. Of these, 6 reached consensus in round two, and 12 in round three. Fifteen elements reached the consensus threshold in total. Of these, eight reached consensus in round two, and seven in round three. Five items did not reach consensus, and six elements did not reach consensus. The Kruskal–Wallis tests revealed that for all but one of the items and elements selected for the tool, the medians were considered equal across the expert groups. The data for each item and element selected for the tool are included in Table 3, while the data for each item and element not selected for the tool are included in Table 4.

**Table 3.** Items and elements selected for the tool.

| Tool Item | Round 2 (% Rated of Critical Importance/Median) | Round 3 (% Rated of Critical Importance/Median) | Kruskal–Wallis Test | Number Panellists Who Rated the Item (in Round Where It Met Consensus Threshold) |
|---|---|---|---|---|
| 1. Opening statement to include: | | | | |
| Description of the nature of the tool itself | 70.9%/7.5 | 86.5%/7.7 | H(2) = .950, *p* = .622 | 52 |
| Family-friendly neglect definition | 84%/7.8 | Not required | H(2) = 3.938, *p* = .140 | 50 |
| Neglect definition 1 * | 68.5%/6.7 | 71.4%/7.1 | H(2) = 4.316, *p* = .116 | 49 |
| Executive summary below the tool's opening statement | 62.5%/7 | 84.6%/7.4 | H(2) = 3.879, *p* = .144 | 52 |
| 2. How to identify neglect in the tool: | | | | |
| List of neglect types | 68.5%/7 | 82.7%/7.4 | H(2) = 2.010, *p* = .366 | 52 |
| 3. How to identify family, organisational, and societal neglect drivers: | | | | |
| Section for each | 73.1%/7.2 | 88.2%/7.7 | H(2) = 0.481, *p* = .786 | 51 |
| Each section to focus on strengths and concerns | 98%/8.5 | Not required | H(2) = 9.002, *p* = .11 (item kept as all expert group means greater than 7) | 51 |
| Each section to focus on dynamic factors | 84.3%/8.1 | Not required | H(2) = 1.343, *p* = .511 | 51 |

**Table 3.** *Cont.*

| Tool Item | Round 2 (% Rated of Critical Importance/Median) | Round 3 (% Rated of Critical Importance/Median) | Kruskal–Wallis Test | Number Panellists Who Rated the Item (in Round Where It Met Consensus Threshold) |
|---|---|---|---|---|
| 4. Tool scales (design): | | | | |
| 10-point scales | 69.2%/7.1 | 86.3%/7.6 | H(2) = .087, *p* = .957 | 51 |
| Text box to explain rating given | 92.2%/8.3 | Not required | H(2) = 1.305, *p* = .521 | 51 |
| Text box to be used to provide neglect examples | 80.4%/7.7 | Not required | H(2) = 0.485, *p* = .785 | 51 |
| Text box to be used to identify knowledge to support rating given | 67.3%/7.2 | 86.8%/7.6 | H(2) = 0.852, *p* = .653 | 53 |
| 5. Tool scales (focus on neglect impacts and care provided): | | | | |
| Current impacts for child | 98%/8.6 | Not required | H(2) = .742, *p* = .690 | 50 |
| Anticipated future impacts | 71.2%/7.1 | 84.6%/7.4 | H(2) = 2.055, *p* = .358 | 52 |
| Current level care provided | 82%/7.7 | Not required | H(2) = .467, *p* = .792 | 50 |
| Tool to capture timing of neglect for child | 76.9%/7.7 | 94.2%/7.8 | H(2) = 1.237, *p* = .539 | 52 |
| 6. Support section of the tool: | | | | |
| Scale family's capacity change with support and resources | 82%/7.7 | Not required | H(2) = 2.881, *p* = .237 | 50 |
| Section for level of intervention recommended | 73.5%/7.3 | 94.2%/7.7 | H(2) = 1.186, *p* = .553 | 52 |
| Section for matching neglect issues with available support | 67.3%/7.3 | 80.8%/7.4 | H(2) = 4.185, *p* = .123 | 52 |
| Section for previous support and its effectiveness | 70%/7.2 | 94.3%/8.1 | H(2) = 1.131, *p* = .568 | 53 |
| Section for parents' aspirations for child | 68%/7 | 83%/7.5 | H(2) = .906, *p* = .636 | 53 |
| Section for follow-up review | 89.6%/8 | Not required | H(2) = 2.386, *p* = .303 | 48 |
| 7. How to best capture parents and carers' views: | | | | |
| Open text box with prompts | 77.6%/7.7 | 96.2%/8.8 | H(2) = 3.079, *p* = .214 | 53 |
| 8. How to best capture children/young people's views: | | | | |
| Open text box with prompts | 79.2%/7.3 | 83%/8.6 | H(2) = 2.595, *p* = .273 | 53 |
| 9. Professionals' contributions to the tool: | | | | |
| One lead professional responsible for tool | 78.4%/7.4 | 90.2%/7.9 | H(2) = 4.119, *p* = .128 | 51 |
| Other professionals to complete only sections relevant to them | 59.2%/6.6 | 73.1%/7 (decision taken to include as this option scored significantly higher than the other option proposed to the panel—please see Table [4]) | H(2) = 1.220, *p* = .543 | 52 |
| 10. Tool to contain hyperlinks to guidance and research for: | | | | |
| Types of neglect | 82%/7.7 | Not required | H(2) = .485, *p* = .785 | 50 |
| Neglect severity and chronicity | 82.4%/7.6 | Not required | H(2) = 2.147, *p* = .342 | 51 |
| Causes and complicating factors for neglect | 84.3%/7.6 | Not required | H(2) = 3.264, *p* = .196 | 51 |
| Impacts for child | 88.2%/8.1 | Not required | H(2) = 3.157, *p* = .206 | 51 |
| Support for family by multi-agency team | 72.6%/7.2 | 92.2%/7.7 | H(2) = 1.756, *p* = .416 | 51 |
| Parent/carer capacity change | 80.4%/7.5 | Not required | H(2) = 8.855, *p* = .012 (item kept as all expert group means greater than 7) | 51 |

**Table 3.** *Cont.*

| Tool Item | Round 2 (% Rated of Critical Importance/Median) | Round 3 (% Rated of Critical Importance/Median) | Kruskal–Wallis Test | Number Panellists Who Rated the Item (in Round Where It Met Consensus Threshold) |
|---|---|---|---|---|
| 11. Guidance for assessors completing the tool: Include how to complete tool, that tool draws on best evidence, and explanation about its focus on how social disadvantages can contribute to neglect | 65.3%/7.1 | 86.3%/7.8 | H(2) = .994, *p* = .608 | 51 |

\* Neglect definition 1: Neglect is when a child's needs are not met, to a level that results in avoidable significant harm to their health, development or wellbeing. Neglect may be caused by family difficulties or through families not having enough resources or support to meet their children's needs.

**Table 4.** Items and elements not selected for the tool.

| Tool Item | Round 2 (% Rated of Critical Importance/Median) | Round 3 (% Rated of Critical Importance/Median) | Number Panellists Who Rated the Item (in Round 3) |
|---|---|---|---|
| 1. Opening statement to include: | | | |
| Emphasis on children's rights | 69.8%/7.5 | 78.4%/7.3 | 51 |
| Neglect definition 2 * | 45.3%/6.2 | 54.9%/6.5 | 51 |
| 2. How to identify neglect in the tool: | | | |
| Open text box with prompts | 59.6%/6.8 | 61.5%/6.6 | 52 |
| 3. How to identify family, organisational, and societal neglect drivers: | | | |
| Open text box with prompts | 63.3%/6.7 | 56%/6.4 | 50 |
| 4. Tool scales (design): | | | |
| Traffic light system | 56.9%/6.8 | 52%/6.2 | 50 |
| 5. How to best capture parents and carers' views: | | | |
| Set questions to ask parent/carer | 57.1%/6.4 | 41.2%/6.2 | 51 |
| 6. How to best capture children/young people's views: | | | |
| Set questions to ask child/young person | 41.7%/6.6 | 31.4%/5.3 | 51 |
| Open text box with prompts and options for drawing by the child/young person | 72.6%/8.6 | 82.3%/8.6 | 52 |
| 7. Professionals' contributions to the tool: | | | |
| Non-lead professionals to complete all sections of tool | 45.7%/6.1 | 40%/5.7 | 50 |
| 8. Tool to contain hyperlinks to guidance and research for: | | | |
| Level of care provided | 78%/7.5 | 78.4%/7.4 | 51 |
| 9. Guidance for assessors completing the tool: | | | |
| Very short and simple, focussing on how to complete tool | 61.2%/6.6 | 42%/5.7 | 50 |
| Include how to complete tool and that tool draws on best evidence | 59.6%/6.7 | 56%/6.6 | 50 |

\* Neglect definition 2: Neglect is when there is an absence of care or resources for a child that results in avoidable significant harm to their health, development, or wellbeing. For the purpose of this assessment, we need to understand if this is a result of parental care or a lack of resources or support being provided for the family by organisations or government. Note: Scales for neglect severity and chronicity universally designated as essential by the panel in round 1, so taken directly to be included in the tool.

There were two items where reaching consensus was more complicated. The panel agreed that the tool should use a family-friendly definition of neglect (a definition that does not pathologise families), but the two options offered in round two did not reach consensus. We, therefore, held a focus group with this study's advisory group, leading to both options being amended for round three. In round three, neither option reached consensus; 71.4% of the panel rated option one as of critical importance, and 54.9% rated option two as of critical importance. We decided to include option one in the tool, as this option scored significantly higher. This defines neglect as 'when a child/young person's needs are not met, to a level that results in avoidable harm to their health, development or wellbeing. Neglect may be caused by family difficulties or through families not having enough resources or support to meet their children's needs'.

Two options for capturing children's and young people's views reached the 80% threshold for inclusion. Option one (an open text box with prompts) was included, as it had a higher rating (83%) than option two (an open text box with prompts and options for drawing by the child/young person) (82.3%). However, given how close these ratings were, an option was included in the tool to attach a drawing.

The panel agreed that questions in the tool using a scale as the answer type should be positively scaled, with scales running from 0 to 10. Furthermore, these should be augmented by qualitative data. So, for example, the tool asks for a numerical rating running from low to high in severity, and then asks for examples of the severity of the neglect. The panel agreed on the importance of including a range of hyperlinks to guidance and research. The aim was to include one short piece of research/guidance for ease in practice and one longer open-access academic journal article to encourage research literacy. However, given the limited research base for child neglect, this was not possible for all options. The hyperlinks will be reviewed annually to ensure the knowledge being accessed is up to date.

The free-text responses of the panellists in the Delphi rounds revealed a number of themes important for the tool and its development. One way the group suggested the tool could support informed practitioner decision-making was by adding free-text boxes linked to scales, with these boxes used to provide evidence to support the rating given. They also suggested that including a section on parents' aspirations for their children could support motivation for change, and further, that the review section of the tool should be set at 3–6 months and used to review the actions taken and the support services offered and their impacts for better or worse.

The draft family and wider social neglect measurement tool was developed from the items and elements that reached consensus in the Delphi study. An outline of its contents can be found in Table 5 on page 19.

**Table 5.** Contents of the 'family and wider social neglect measurement tool'.

| Section | Focus of the Section |
|---|---|
| 1. Introduction to the tool | • Tool's ethos of being family-centred and completed with families to assess both strengths and concerns.<br>• Tool's aims of balanced and evidence-informed assessments that are inclusive of social harms and supportive of proactive and preventative practice.<br>• Neglect definition adopted.<br>• Assessment overview box to capture the key points of the completed assessment. |
| 2. Current level of care and how severe and chronic the neglect is | • Scales for current level of care, neglect severity, and how chronic the neglect is.<br>• Accompanying free-text boxes for current level of care, neglect severity, and chronicity that ask the assessor to provide examples supporting the rating and key evidence from research or guidance supporting the rating.<br>• Hyperlinks to research on neglect severity and chronicity. |

**Table 5.** *Cont.*

| Section | Focus of the Section |
|---|---|
| 3. Neglect identification | <ul><li>Asks assessor to identify which neglects from the tool's neglect typology are present (physical, medical, educational, emotional, social, and lack of supervision and guidance).</li><li>Asks assessor to identify the severity of each neglect type—mild, moderate, or severe.</li><li>Hyperlink to research on types of neglect.</li></ul> |
| 4. Impacts of neglect for the child/young person | <ul><li>Scales for current and anticipated future impacts of the neglect for the child/young person.</li><li>Accompanying free-text boxes that ask the assessor to provide examples supporting the rating and key evidence from research or guidance supporting the rating.</li><li>Asks assessor to evaluate the timing of the neglect for the child/young person and the significance of the timing of the neglect for the child/young person and their development.</li></ul> |
| 5. Causes, complicating factors, and strengths | <ul><li>Focusses on causes, complicating factors, and strengths at the family, organisational, and community/society levels.</li><li>For each, asks the assessor to identify concerns, strengths, and dynamic factors (factors open to change).</li><li>Hyperlink to research on causes and complicating factors for neglect.</li></ul> |
| 6. Family members' views | <ul><li>Asks for accurate and full account of parents/carers' and child/young person's views on family life, levels of care, neglect concerns, strengths, and support they need.</li><li>Asks for parents/carers' hopes and aspirations for the child/young person and how these can be used to encourage positive change.</li></ul> |
| 7. Support for the family | <ul><li>Focusses on support and change at family, community, and society levels.</li><li>Scale for family's capacity to address the neglect concerns with appropriate support and resources.</li><li>Accompanying free-text box that asks the assessor for examples supporting the rating and key evidence from research or guidance supporting the rating.</li><li>Asks the assessor to match key issues and neglect causes with available support and services to develop the support plan.</li><li>Hyperlinks to research on capacity for change and support for families.</li></ul> |
| 8. Summary of scores and level of intervention | <ul><li>Summary of scores for completed scales.</li><li>Asks for level of intervention recommended.</li><li>Provides guidance on levels of intervention for mild, moderate, and severe neglect.</li></ul> |
| 9. Follow-up review | <ul><li>Review at 3/6 months, including on main neglect concerns, causes/contributing factors, strengths, support provided, and level of intervention recommended.</li><li>Support plan for next 3 or 6 months.</li></ul> |
| Guidance for assessors | <ul><li>Concise guidance on how to complete the tool as a multi-agency team, how to draw on best evidence, and the ways social disadvantages can contribute to neglect.</li><li>Hyperlink to national guidance on child neglect.</li></ul> |

## 4. Discussion

This study represents the first effort in the field of social work to identify and reach expert consensus through a Delphi study on the development of a new child neglect measurement tool. The draft 'family and wider social neglect measurement tool' was developed through a rigorous and systematic but also collaborative research project. The

overarching evidence-based methodology has been inclusive of the knowledge developed through practice and lived experience, in line with more recent trends in evidence-based research (Oliver et al. 2019; Wieringa and Greenhalgh 2015). This has been important for a research project focussed on impacting real-world practice and producing knowledge in ethical and fair ways (Barber et al. 2011). The Delphi study reported in this paper has acted as an important stage in this process, building on the findings of the systematic review to develop items and elements for the draft tool. It offered a systematic approach to gathering the views of a range of experts, free from group pressures and associated socio-cognitive biases (Grant et al. 2018; Khodyakov et al. 2020). Delphi studies can act as important components of evidence-based approaches in under-researched areas, such as child neglect (Lee et al. 2011).

There are a number of features of the Delphi study that have promoted the draft tool's internal, content, and construct validities and sensitivity and specificity. The application of the CREDES guidelines supported a comprehensive and rigorous Delphi study. The Delphi panel constituted a relatively large number of experts in child neglect. There were very high response rates and high rates of agreement among the panel members and the three different expert groups as to what items and elements should be included in the tool. We set an 80% consensus threshold for the inclusion of items and elements.

Proctor and Dubowitz (2014) stated 'At a minimum, an assessment should determine whether or not neglect has occurred, the nature and severity of the neglect, whether the child will be safe, what factors are contributing to the neglect, what protective factors are present, and what interventions have been tried, with what results' (p. 44). Our draft tool covers these fundamental areas required to be comprehensive for assessing child neglect, and may offer face, content, and construct validity. Its reliability, validity, sensitivity, and specificity will need to be tested in the forthcoming pilot phase of the project.

The included items and elements should support the tool's aims of:

- Accurately assessing child neglect;
- Supporting balanced and evidence-informed assessments inclusive of strengths as well as concerns;
- Supporting assessments inclusive of factors that make family life and family wellbeing harder, such as social isolation and poor housing.

The draft tool has nine short-labelled sections and clear, concise guidance for assessors. It contains hyperlinks to research and guidance on key areas, such as neglect severity, the adopted neglect typology, and causes of and complicating factors for neglect. Table 5 on page 19 shows the main contents of the tool.

There are a number of features that distinguish the family and wider social neglect measurement tool from other child neglect assessment tools we examined in the systematic review and commonly used tools such as the Graded Care Profile (1 and 2). Although these tools have strengths and important features to learn from, they all present missing elements. Firstly, our tool is free for all, not behind a paywall. It focusses on the presence or absence of actual child neglect, and its severity, chronicity, and type. This differs from tools such as the Graded Care Profile (1 and 2) or the HOME tool, which essentially assess the quality of care provided. There is a range of differences between our tool and those considered to have been rigorously tested in the systematic review. It adopts the comprehensive neglect typology used for this study, assesses neglect chronicity, has a specific support section to indicate the type of support the family requires, and incorporates a review section to measure change. It can complement more general children and families in multi-agency assessments.

The adoption of a social harm framework in the project and Delphi study offers a new approach to understanding child neglect within the contexts of wider society, government policies, and organisational practices. It provides a robust lens for analysing the complex drivers for neglect and family (dys)function from family to societal levels. There have, therefore, been fundamentally different conceptual and value bases guiding the tool's development. In the Delphi study, the panellists were asked to consider the relevance of

social harms to the tool and how these could be included in the tool. Other tools, such as the GCP (1 and 2) and Trocmé's (1996) Child Neglect Index, for example, primarily focus on the family level, whereas the family and wider social neglect measurement tool, supported by the social harm framework, looks to key risk and protective factors for neglect from the family to societal levels, while having an ethos of being family-focussed and not just child-focussed.

However, this study has limitations. The results offer the collective views of a particular group of experts on measuring child neglect (Hasson and Keeney 2011). The Delphi panellists were mainly from the UK, and a majority were White British practitioners. Experiential and practitioner knowledge has been criticised for simply reflecting their own experiences and outlooks, while lacking a wider understanding of the systems and societies in which they work or participate (Castro et al. 2018; Solbjør and Steinsbekk 2011). This was evident in some of the free-text answers and suggestions for the key drivers of neglect. While Delphi studies are viewed as democratic processes, those in the minority groups (experts by experience and academics) may have been influenced to change their views based on the views of those in the majority group (practitioners) (Powell 2003). There remains limited guidance on the desired balance between qualitative and quantitative data in Delphi studies. The approach used in this study may have differed from another group of researchers approaching the same study, leading to potentially different results and, therefore, a different tool (Keeney et al. 2001). The draft tool remains to be tested, but this is currently underway in a pilot study as the final stage of this project.

### 4.1. Implications for Practice

The family and wider social neglect tool aims to support evidence-based and research-informed child neglect assessments and decision-making in practice. This is important given that child welfare academics have, over many years, advocated for more research and evidence-based approaches to assessing child neglect (Brandon et al. 2013; Dubowitz et al. 1993; Dubowitz 2007; Macdonald 2001; Stevenson 1998; Tanner and Turney 2003; Daniel 2015). The social harm framework adopted and enacted in the Delphi study reminds practitioners that neglect cases are often characterised by difficulties ranging from the familial to the societal level and families facing a range of social harms, notably, socioeconomic disadvantage (Bywaters et al. 2022; Lacharité 2014). The tool's focus on strengths and concerns, alongside this range of drivers for neglect, should encourage family-centred practice and a focus on needs and unmet needs, as opposed to a singular focus on risk. The inclusion of parental hopes and aspirations for their children reflects the literature that suggests their importance for motivation to change (Boddy et al. 2016; Koprowska 2014). The tool's focus on community-based support may act as one step towards reconnecting professional systems with communities and the support they can offer.

Research on the impacts of COVID-19 on practice has revealed new time pressures on social workers and allied professionals and less opportunity for them to visit families to assess family life and environments (Baginsky and Manthorpe 2020; Cook and Zschomler 2020; Ferguson et al. 2022). Our succinct tool should support practitioners to produce concise neglect-focussed assessments within this new practice landscape.

### 4.2. Implications for Research

This study has demonstrated the potential benefits of employing the Delphi technique for the development of tools and measures in social work research. The study design, with distinct developmental stages followed by Delphi rounds, can function as one example for the development of rigorous Delphi approaches in social work research. The approach adopted has shown how Delphi studies and evidence-based approaches can be inclusive, collaborative, and ethical while generating robust and valid knowledge.

The inclusion of parents with experience with children and family social work (an often-marginalised group in research and practice) in a Delphi study and project focussed on a statutory multi-agency arena demonstrates some possibilities for evidence-based research

that aims to be inclusive of the knowledge gained through lived experience. The approach adopted chimes with codes of ethics for social work research, such as those presented by Butler (2002) and JUCSWEC (Joint University Council Social Work Education Committee) (2008). The focus on neglect as a social form of harm with a range of drivers is important for research that aims to study neglect within its wider social and societal contexts.

*4.3. Next Steps in the Study*

The next phase of this project is to pilot the draft child neglect measurement tool with multi-agency practitioners in the participating local authority. This phase will test the tool's validity, reliability, sensitivity, specificity, and useability in practice. It will employ a test-retest method and gain the views of practitioners and families on the tool.

## 5. Conclusions

This Delphi study employed a mixed panel of experts to develop a new multi-agency child neglect measurement tool. The tool is succinct, may be useable by a range of practitioners in multi-agency settings, and is inclusive of how social harms can contribute to neglect. It aims to support informed assessments and decision-making in cases of child neglect.

**Author Contributions:** Conceptualization, S.H., P.M. and J.S.; methodology, S.H., P.M. and J.S.; formal analysis, S.H.; investigation, S.H.; data curation, S.H.; writing—original draft preparation, S.H.; writing—review and editing, S.H., P.M. and J.S. All authors have read and agreed to the published version of the manuscript.

**Funding:** This research was partially funded by Neath Port Talbot. Agreement number: 1941869.

**Institutional Review Board Statement:** Application for Ethical Review ERN_21-0041: Thank you for your application for ethical review for the above project, which was reviewed by the Humanities and Social Sciences Ethical Review Committee of the University of Birmingham. On behalf of the Committee, I confirm that this study now has full ethical approval.

**Informed Consent Statement:** Informed consent was obtained from all subjects involved in the study.

**Data Availability Statement:** The data presented in this study are available on request from the corresponding author, Simon Haworth. The data are not publicly available due to privacy and ethical restrictions.

**Acknowledgments:** We would like to thank the participating local authority Neath Port Talbot. We would also like to thank the experts by experience, practitioners, and academics who kindly contributed to the focus groups and constituted the Delphi panel.

**Conflicts of Interest:** The authors declare no conflict of interest.

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
