# Peer review of "A Delphi Study to Develop Items for a New Tool for Measuring Child Neglect for Use by Multi-Agency Practitioners in the UK"

_socsci, doi:10.3390/socsci12040239_

Round 1

Reviewer 1 Report

This paper describes a Delphi study that has been undergone in order to develop a new tool measuring child neglect for use by multi-agency practicioners. The Delphi method seems to have been followed with rigor and methodological choices are justified. The quality of the writing is good. There is potential for significant contribution of this paper, but major issues of clarity and coherence must be addressed before.

Throughout the paper, the objectives of the study are unclear. According to Section 1.3 The Research Project (p.2, lines 71-74), the overarching aim of the paper to develop a new measurement tool for child neglect, using a collaborative approach, relying on a home-made theory of child neglect and the Social Harm model (more specific objectives are not provided). However, the results presented are derived exclusively from the Delphi process, which is the 2nd of three steps to develop the tool (the first being the focus groups and the 3rd being the pilot testing). The methodology and the results make it possible to identify the contours of the tool to be developed, since this seems to have been the purpose of the Delphi surveys, but certainly not to have a clear and concrete idea of the items that will compose the tool and the construct that will be measured. However, the discussion is formulated as if the Delphi process, as itself, had allowed the tool to be built and validated. The author seems to claim that the tool developed through the Delphi surveys offers validity, sensitivity and specificity, whereas the study design presented does not refer to any measure of validity.

It is difficult to appreciate the real contribution of the article in its current version, because the results presented seem to be interim and incomplete with respect to the objective set. The article does not give the reader a clear idea of what constructs are being measured, how they will be measured, neither how the Delphi process could have contribute to the development of the final tool.

I recommend clarifying the specific objectives of the article (not just the general purpose of the study it is part of) and discussing the results in relation to those objectives.

Author Response

Dear Reviewer 1

Thank you for your careful consideration of our paper. Please see below our responses to your comments and the associated changes we have made to our paper.

Reviewer 1:

Comments and Suggestions for Authors

This paper describes a Delphi study that has been undergone in order to develop a new tool measuring child neglect for use by multi-agency practitioners. The Delphi method seems to have been followed with rigor and methodological choices are justified. The quality of the writing is good. There is potential for significant contribution of this paper, but major issues of clarity and coherence must be addressed before.

  1. Throughout the paper, the objectives of the study are unclear. According to Section 3 The Research Project(p.2, lines 71-74), the overarching aim of the paper to develop a new measurement tool for child neglect, using a collaborative approach, relying on a home-made theory of child neglect and the Social Harm model (more specific objectives are not provided). However, the results presented are derived exclusively from the Delphi process, which is the 2nd of three steps to develop the tool (the first being the focus groups and the 3rd being the pilot testing).

Response:

Thank you for this useful observation. We have now amended Section 1.3 – The research project (p.3, lines 116-117) to clearly communicate at its end that this paper reports the Delphi study phase only.

It now states: This paper reports the Delphi study phase of the project. This was employed to develop items and elements for the draft tool.

We have amended the start of Section 4- Discussion (p.19, lines 398-402) to make clearer that the paper is focussed on the Delphi and its results, which feed into the overarching research project through developing key items and elements for the draft tool.

This now states: The Delphi study reported in this paper has acted as an important stage in this process, building on the findings of the systematic review to develop items and elements for the draft tool. Delphi studies can act as important components of evidence-based approaches in under researched areas, such as child neglect (Lee et al., 2011).

We have also amended Section 4 – Discussion to include how the social harm framework informed the Delphi study.

This now states (p.22, lines 442-44): The adoption of a social harm framework in the project and Delphi study has offered a new approach for understanding child neglect within the contexts of wider society, government policies and organisational practices.

This now states (p.23, lines 447-448): In the Delphi study, panellists were asked to consider the relevance of social harms to the tool and how these could be included in the tool.

In Section 4.1 – Implications for practice (p.23, lines 474-478), it now states: The social harm framework adopted, and enacted in the Delphi study, reminds practitioners that neglect cases are often characterised by difficulties ranging from the familial to the societal level and families facing a range of social harms, notably socioeconomic disadvantage (Bywaters et al., 2022; Lacharite, 2014).

  1. The methodology and the results make it possible to identify the contours of the tool to be developed, since this seems to have been the purpose of the Delphi surveys, but certainly not to have a clear and concrete idea of the items that will compose the tool and the construct that will be measured.

Response:

Thank you for this important observation. We have now developed Table 5 to offer a more comprehensive overview of the tool and what it measures. We have moved this table to the end of the Results section to clarify the results of the Delphi study and complement discussion of the results of the Delphi study. It can now be found on page 19.

We have also stated more clearly the child neglect definition adopted by the tool in the discussion of the results (Section 3.2 – Delphi study, p. 9, lines 352-355), to offer clarity on how the overall construct of neglect is defined in the tool itself.

The addition states: This defines neglect as: ‘Neglect is when a child/young person's needs are not met, to a level that results in avoidable harm to their health, development or wellbeing. Neglect may be caused by family difficulties or through families not having enough resources or support to meet their children’s needs’.

  1. However, the discussion is formulated as if the Delphi process, as itself, had allowed the tool to be built and validated. The author seems to claim that the tool developed through the Delphi surveys offers validity, sensitivity and specificity, whereas the study design presented does not refer to any measure of validity.

Response:

Thank you for this observation. We have amended Section 4 - Discussion (p.21, lines 413-416) to more clearly describe that we have developed a draft tool through the Delphi that may offer content and face validity, but that further testing is required.

It now states: Our draft tool covers these fundamental areas required to be comprehensive for assessing child neglect, and may offer face, content and construct validity. Reliability, validity, sensitivity and specificity will need to be tested in the forthcoming pilot phase of the project.

Further, we have more clearly reiterated the Delphi study’s role in the overall research project in Section 4 - Discussion, i.e. developing items and elements for the draft tool as described above.

  1. It is difficult to appreciate the real contribution of the article in its current version, because the results presented seem to be interim and incomplete with respect to the objective set. The article does not give the reader a clear idea of what constructs are being measured, how they will be measured, neither how the Delphi process could have contribute to the development of the final tool.

Response:

Thank you for this interesting comment. We have now amended Section 1.3 – The research project, alongside both the Results and Discussion sections to more clearly describe the importance of the Delphi study in the overall project, the draft nature of the tool and the contents of the tool itself. It is a critical part in the process and we believe that this paper illuminates both the technique and the results themselves for readers.

  1. I recommend clarifying the specific objectives of the article (not just the general purpose of the study it is part of) and discussing the results in relation to those objectives.

Response:

Thank you for this observation. As described above we have now modified our manuscript to provide greater clarity on its specific objectives: (Section 1.3 – The research project), its focus on the Delphi phase (Sections 1.3 and Section 4), and the results (Section 3) in relation to these objectives.

Reviewer 2 Report

The manuscript addresses an important and understudied topic of research. However, minor revisions are recommended prior to publication. Details regarding my specific proposed changes are outlined below.

Introduction

1.     Missing from the Introduction section is a discussion of how definitions of neglect vary across jurisdictions. Different conceptual models of neglect as well as different neglect typologies also exist which have further limited efforts to measure and assess child neglect. Adding a brief description of these issues would help the reader to understand the full range of factors that have undermined progress in the development of neglect assessment tools.

2.       One strength of this manuscript is the inclusion of the child neglect theory of change which served as a framework to guide the project. The inclusion of both risk and protective factors at multiple levels of the child’s social ecology (i.e., personal, family, professional, community and societal levels) is well aligned with previous research and with developmental theory. However, it is unclear why the model is called “theory of change” as change does not appear to be a central component of the discussion of child neglect in this manuscript. In addition, one improvement of the model would be to add a “family” risk factors box to Figure 1 to incorporate known risk factors for neglect that operate at the family-level, such as challenges with securing reliable childcare and family social support. Perhaps the current “absence of support and social connections” box could be incorporated into the “parental” box if these associations are specific to the parents. Alternatively, if the proposed conceptual model suggests these factors operate at the family level, then perhaps this content could be added to a family-level box.

3.       Also consider adding “disability” to the child vulnerabilities box based on the growing literature supporting the increased risk of neglect among children who have documented disabilities.

4.       Another meaningful expansion of the model would be to depict how the identified risk and protective factors are differentially associated with the neglect types listed in the middle green boxes.

5.       Page 2 – line 56. Elaboration of the agencies typically involved in assessing child neglect would be helpful here.

Method

1.       Page 5, line 170 – Is it possible to differentiate or clarify whether these views are those of the authors or are empirically supported based on research that has utilized the Modified Maltreatment Classification System?

2.       Page 5, line 176 – More details regarding which gold standard is referenced here would be helpful.

3.       Page 6, line 257 – Data from the Delphi rounds were described as being analyzed using “qualitative content analysis”. What specific software or coding program was used to analyze these data? Methods and therefore results can vary widely within this broad description.

4.       Page 8, Table 1 – It would be helpful to know the professions of the 3 “other professionals” who participated in the focus groups given the small sample size. If this information is too cumbersome to include in the table, perhaps it could be added to the text.

5.       Page 9, Line 321-324 – Please clarify the difference between items and elements. Without clarification, it is challenging to map the description of Table 3 in this section onto the actual content in Table 3.

6.       Page 19,  Line 394-398 – The description of the tool beginning on this line (including Table 5) would be better placed in the Results section to clarify the outcome of the Delphi process.

7.       Table 5 – What is the rationale for placing the timing of neglect in Section 4 which concerns the impact of neglect? The timing of neglect may be better placed in Section 3 (Neglect Identification).

8.       It is unclear how the caregiver’s hopes and aspirations of the child are central to the goal of assessing whether child neglect occurred.

Results and Discussion

1.       What is the rationale for not including the items from the proposed measurement tool in this manuscript? Is the intention to publish this information in a subsequent paper on this tool (the pilot study)? As a reader and an academic with expertise on child neglect, I was expecting to see items full to be included in the tool provided in this manuscript. Without providing the actual items included in the proposed tool, it is not possible to comment on the internal and construct validity of the proposed tool. It in addition, it is not possible to evaluate comments in the Discussion section about the comprehensiveness of the risk factors assessed in the tool (e.g., “It provides a robust lens for analysing the complex drivers for neglect and family (dys)functioning from family to through (sic) to societal levels.”)

2.       On line 471, the tool is finally described as in “draft” form.  If this is the justification for why the tool is not fully produced in this publication, then consider introducing this description early in the manuscript. 

3.       Although the authors argue that the tool is family focused, family is not one of the domains of risk factors depicted in Figure 1. The coverage of family-level characteristics as a protective factor is minimal.

Author Response

Dear Reviewer 2

Thank you for your careful consideration of our paper. Please see below our responses to your comments and the associated changes we have made to our paper.

Reviewer 2:

Comments and suggestions for authors

The manuscript addresses an important and understudied topic of research. However, minor revisions are recommended prior to publication. Details regarding my specific proposed changes are outlined below.

Thank you for these supportive remarks.

Introduction

  1. Missing from the Introduction section is a discussion of how definitions of neglect vary across jurisdictions. Different conceptual models of neglect as well as different neglect typologies also exist which have further limited efforts to measure and assess child neglect. Adding a brief description of these issues would help the reader to understand the full range of factors that have undermined progress in the development of neglect assessment tools.

Response:

Thank you for this constructive comment. We agree completely and have added a short section focussed on issues in definitions, typologies and conceptual models to section 1.1 (pp.1-2, lines 42-46).

This now states: There are a significant range of definitions of child neglect from research, government, and practice (English et al., 2005). Definitions vary between countries and indeed between states and jurisdictions within countries (Horwath, 2013). There are also a range of conceptual models and typologies of child neglect (Horwath, 2007; Sullivan, 2000). These issues create a complex picture for assessment.

  1. One strength of this manuscript is the inclusion of the child neglect theory of change which served as a framework to guide the project. The inclusion of both risk and protective factors at multiple levels of the child’s social ecology (i.e., personal, family, professional, community and societal levels) is well aligned with previous research and with developmental theory. However, it is unclear why the model is called “theory of change” as change does not appear to be a central component of the discussion of child neglect in this manuscript. In addition, one improvement of the model would be to add a “family” risk factors box to Figure 1 to incorporate known risk factors for neglect that operate at the family-level, such as challenges with securing reliable childcare and family social support. Perhaps the current “absence of support and social connections” box could be incorporated into the “parental” box if these associations are specific to the parents. Alternatively, if the proposed conceptual model suggests these factors operate at the family level, then perhaps this content could be added to a family-level box.

Response:

Thank you, this is a point well raised. We have relabelled the boxes as ‘Parental and family issues’ and ‘Parental strengths and positive family functioning’ to reflect your suggestions. We have not wanted to significantly change the theory of change as it has been developed from the literature and consultation with stakeholders.

We have slightly amended its explanation in Section 1.3 – The research project (p.3, lines 97-98). This now states: Developed from a review of the literature on neglect, its key dimensions and drivers, and children’s needs, this study’s neglect typology and consultation with our advisory group, our child neglect theory of change (see fig 1 below) provides a framework to guide the project.

We have labelled this as a theory of change as it describes and explains the complex social mechanisms contributing to child neglect. We recognise that it does not fit a ‘traditional’ theory of change approach focussed on the project’s aims and outcomes. However, it does look to clearly communicate the project’s central guiding framework and how these complex social mechanisms can impact changes upon children at risk of neglect and their needs- a process of change in itself.

  1. Also consider adding “disability” to the child vulnerabilities box based on the growing literature supporting the increased risk of neglect among children who have documented disabilities.

Response:

Agreed-  we have clarified this and added this to the child vulnerabilities box.

  1. Another meaningful expansion of the model would be to depict how the identified risk and protective factors are differentially associated with the neglect types listed in the middle green boxes.

Response:

Thank you for this idea. We have given it full consideration, but believe that one strength of the theory of change is its visual simplicity. Therefore, although the idea has clear merits, we want to avoid adding complexity to the theory of change. The data we currently have does not support this level of detail.

  1. Page 2 – line 56.Elaboration of the agencies typically involved in assessing child neglect would be helpful here.

Response:

Thank you for your observation. We have elaborated the agencies typically involved to Section 1.2 – Assessment challenges (p.2, lines 66-67).

This now states: Health, early help and education agencies are commonly involved in assessing and responding to child neglect (Sharley, 2020).

Method

  1. Page 5, line 170 – Is it possible to differentiate or clarify whether these views are those of the authors or are empirically supported based on research that has utilized the Modified Maltreatment Classification System?

Response:

Thank you for this observation. In Section 2.2.1 – Systematic review (p.5, lines 177-180), we have clarified that this view is based on both our analysis and the study by Dubowitz et al. (2005) examining the Modified Maltreatment Classification System. 

This now states: Only one tool, the Child Neglect Index (Trocme, 1996), was considered feasible for practice, with the modifications of the Modified Maltreatment Classification System considered too complicated and cumbersome in both our review and the study of Dubowitz et al. (2005) examining the tool.

  1. Page 5, line 176 – More details regarding which gold standard is referenced here would be helpful.

Response:

Thank you for this comment. We have amended Section 2.2.1 – Systematic review (p.5, lines 183-186), to provide more detail on the gold standard.

This now states: Studies of ‘popular’ tools such as the Graded Care Profile 2 and HOME lacked methodological rigour and had not been assessed against the gold standard of a contemporaneous assessment by a qualified children’s social worker or by an assessor working within children’s social work (Author 1 et al., 2022).

  1. Page 6, line 257 – Data from the Delphi rounds were described as being analyzed using “qualitative content analysis”. What specific software or coding program was used to analyze these data? Methods and therefore results can vary widely within this broad description.

Response:

Thank you for your observation. We have added detail to our description of the qualitative content analysis in Section 2.4.1 - Qualitative analysis (p.7, lines 270-274).

      This now states: As the data set was relatively small, manual coding was undertaken, supported by the Qualtrics platform to count the categories that emerged from the data (Chew & Eysenbach, 2010). Those most frequently present were then taken forwards as key concepts for panellists to consider in the next round.

  1. Page 8, Table 1 – It would be helpful to know the professions of the 3 “other professionals” who participated in the focus groups given the small sample size. If this information is too cumbersome to include in the table, perhaps it could be added to the text.

Response:

Thank you for your comment. We have added this to the text in Section 3.1 – Focus groups (p.7, lines 304-306).

This now states: Seven of the participants were experts by experience, seven practitioners (from social work, health, education and family support fields) and two academics.

  1. Page 9, Line 321-324 – Please clarify the difference between items and elements. Without clarification, it is challenging to map the description of Table 3 in this section onto the actual content in Table 3.

Response:

Thank you for this observation. We have modified their description in the tool to promote greater clarity.

It now states in Section 2.3.1 – The Delphi rounds (p.6, lines 232-236): The panel rated 45 items (distinct parts for the tool that constitute what the tool assesses and focusses on, for example a scale for neglect severity and the neglect definition used) and elements (features of the tool’s design and look that support it’s aims, for example hyperlinks to research and use of 10 point scales) for the tool in round two on 9-point Likert scales.

For clarity we have now repeated this distinction on page 9 (lines 334-338.).

This now states: Table 3 shows the items (distinct parts for the tool that constitute what the tool assesses and focusses on, for example a scale for neglect severity and the neglect definition used)  and elements (features of the tool’s design and look that support it’s aims, for example hyperlinks to research and use of 10 point scales) that reached consensus to be included in the draft tool.

  1. Page 19,  Line 394-398 – The description of the tool beginning on this line (including Table 5) would be better placed in the Results section to clarify the outcome of the Delphi process.

Response:

Thank you for your observation. We have moved the description and table to the end of the Results section on page 19.

  1. Table 5 – What is the rationale for placing the timing of neglect in Section 4 which concerns the impact of neglect? The timing of neglect may be better placed in Section 3 (Neglect Identification).

Response:

Thank you for this observation. The rationale is that the short timing section in the tool focusses on the developmental phases during which the neglect has occurred and asks practitioners to identify the significance of the timing of the neglect to the child’s development. In this manner linking timing with impacts.

  1. It is unclear how the caregiver’s hopes and aspirations of the child are central to the goal of assessing whether child neglect occurred.

Response:

Thank you for this observation. The inclusion forms part of a balanced approach to assessment. The caregiver’s hopes and aspirations are explored to then encourage positive change. This is briefly described in lines 375-376 on page 10 where it states: They also suggested that including a section on parents’ aspirations for their children could support motivation for change.

We have also now added this to table 5. Further, we have linked this to the literature on the importance of hope and aspiration to change in social work in Section 4.1 – Implications for practice (p.20, lines 480-482).

The addition states: The inclusion of parental hopes and aspirations for their children reflects the literature that suggests their importance for motivation to change (Boddy et al., 2018; Koprowska, 2014).

Results and Discussion

  1. What is the rationale for not including the items from the proposed measurement tool in this manuscript? Is the intention to publish this information in a subsequent paper on this tool (the pilot study)? As a reader and an academic with expertise on child neglect, I was expecting to see items full to be included in the tool provided in this manuscript. Without providing the actual items included in the proposed tool, it is not possible to comment on the internal and construct validity of the proposed tool. It in addition, it is not possible to evaluate comments in the Discussion section about the comprehensiveness of the risk factors assessed in the tool (e.g., “It provides a robust lens for analysing the complex drivers for neglect and family (dys)functioning from family to through (sic) to societal levels.”)

  1. On line 471, the tool is finally described as in “draft” form.  If this is the justification for why the tool is not fully produced in this publication, then consider introducing this description early in the manuscript. 

Response:

Thank you for both of these useful observations. The tool remains in draft form, as we are finding modifications to it currently as we are undertaking the pilot phase, so we elected not to fully produce the tool in this journal article. Rather it will be produced in full in a further journal article reporting the pilot study. We have now amended the manuscript to provide clarity of the tool’s draft status at an earlier juncture. This revision includes the abstract and Section 1.3 – The research project page 2 line 90. Further on page 9 line 338 and page 21 line 413.

We have also developed Table 5 to provide greater detail on the contents of the tool. It can now be found on page 19.

  1. Although the authors argue that the tool is family focused, family is not one of the domains of risk factors depicted in Figure 1. The coverage of family-level characteristics as a protective factor is minimal.

Response:

Thank you for a very useful observation. As described in your point 2, we have relabelled the boxes as ‘Parental and family issues’ and ‘Parental strengths and positive family functioning’ to reflect your suggestions.

Round 2

Author Response

Dear Reviewer 2

Thank you for your careful and considered reviewing of our paper. Please see below our responses to your comments and the associated changes we have made to our paper.

There are still some additional minor corrections that must be made to ensure that the comments are applied consistently throughout the paper. As requested, this version of the manuscript is more focused on the Delphi phase of the overall project and the specific objective of reporting on the Delphi process is clearly stated.

  1. However, with this objective now narrowed, the authors should explain concretely how the process and findings from the Delphi phase can be relevant in themselves, aside from informing the last stage of the overall project (the pilot testing). Some relevance is stated in the discussion section, but the reader should understand right from the introduction of the manuscript, the potential contribution of the paper (for example just after line 116).

Thank you for this constructive comment. We have now amended the sentence after line 116 on page 3 (now lines 119-121). This now reads: This paper reports the Delphi study phase of the project. This was employed to develop items and elements for the draft tool, building on the findings of the preceding systematic review within the overarching evidence-based project. It offered a systematic and efficient approach for gathering of the views of a range of experts (Khodyakov et al., 2020).

We have also amended the discussion section (p.21, lines 408-414) to read: The Delphi study reported in this paper has acted as an important stage in this process, building on the findings of the systematic review to develop items and elements for the draft tool. It offered a systematic approach to gathering the views of a range of experts, free from group pressures and associated socio-cognitive biases (Grant et al., 2018; Khodyakov et al., 2020). Delphi studies can act as important components of evidence-based approaches in under researched areas, such as child neglect (Lee et al., 2011).

  1. The authors need to be more consistent about saying that the tool is to be developed. At several places in the discussion, there are statements that suggest the tool is already finalized and validated thanks to the Delphi study. The contribution of this paper should not be the full development of a new tool. For example:

Line 392: “The ‘family and wider social neglect measurement tool’ has been developed through a rigorous and systematic, but also collaborative research project.”

Line 403: “There are a number features of the Delphi study that have promoted the tool’s internal, content and construct validities, sensitivity and specificity.”

Line 423: “The tool has nine short labelled sections and clear, concise guidance for assessors.”

Response

Thank you for your comment. We have now added the word draft to the three sentences you have highlighted above and reviewed the rest of the manuscript to ensure this is consistent throughout.

  1. In the abstract, the systematic review is presented as the first stage of the Delphi phase, whereas in the manuscript it’s presented as the first phase of the overall project. Please clarify.

Response

Thank you for pointing this out. We have amended the abstract (p.1, lines 7-10) to read: There were two important stages to inform the Delphi study: A systematic review of child neglect measures and three online focus groups with a purposive sample of 16 participants with expertise in child neglect (academics, practitioners and experts by experience).

  1. Lines 97-99, please revise this sentence.

Response:

Thank you for your comment. We have now amended this sentence to ensure it is more clearly written.

It now reads: Our child neglect theory of change (see fig 1 below) provides a framework to guide the project. It has been developed from a review of the literature on neglect (including its key dimensions and drivers) and the literature on children’s needs, alongside consultation with our advisory group.

  1. Tables 4 and 5 should be called in the text before they appear.

Response:

Thank you, this is a point well raised. We have now called tables 4 and 5 in the text.

On page 9 lines 348-350 it now states: The data for each item and element selected for the tool are included in Table 3, while the data for each item and element not selected for the tool are included in Table 4. 

On page 10 lines 385-387 it now states: The draft family and wider social neglect measurement tool has been developed from the items and elements that reached consensus in the Delphi study. An outline of its contents can be found in Table 5 on page 19.